

# Tracking the influence of cloud condensation nuclei on summer diurnal precipitating systems over complex topography in Taiwan

Yu-Hung Chang[1], Wei-Ting Chen[1], Chien-Ming Wu[1], Christopher Moseley[1], Chia-Chun Wu[2]

[1]Department of Atmospheric Sciences, National Taiwan University, Taipei, Taiwan
5  [2]National Science and Technology Center for Disaster Reduction, New Taipei City, Taiwan

*Correspondence to*: Wei-Ting Chen (weitingc@ntu.edu.tw)

**Abstract.** This study focuses on how aerosols, serving as cloud condensation nuclei (CCN), affect the properties of the summertime diurnal precipitation under the weak synoptic weather regime over complex topography in Taiwan. Semi-realistic large-eddy simulations (LESs) were carried out using the vector vorticity equation model with high-resolution 10  Taiwan topography (TaiwanVVM) and driven by idealized observational soundings. Since the aerosol effects on convection could be specific during different stages of the life cycle, we perform object-based tracking analyses, which diagnose both the spatial and temporal connectivity of convective systems, to highlight the convective clouds that are locked by topography and reduce the stochastic features of convection. The statistical analyses on the tracked extreme convective systems highlight the differences in structural characteristics of convection between the experiments with clean and normal CCN scenarios. 15  For the orographic-locking regime, the effects of CCN on the diurnal precipitating systems are more significant. The precipitation initiation is postponed significantly, which prolongs the development of local circulation and convection. The occurrence of the cloud objects with extreme maximum rain rates doubles. Also, the $P_{99}$ of the maximum rain rate and the maximum cloud size during the lifetime of the diurnal precipitating systems increase by 16.9 % and 6.7 %, respectively. This study demonstrates that the object-based tracking analyses of extreme precipitating systems are useful to investigate the 20  responses of orographic-driven diurnal convection to CCN.

## 1 Introduction

Aerosol–cloud–precipitation interactions (ACPI) has been studied extensively in the past few decades to understand how aerosols influence clouds and precipitation through modifying the microphysical processes. Excessive aerosols, released to the atmosphere by continuing human activities, could reshape clouds and precipitation characteristics after being activated as 25  cloud condensation nuclei (CCN). Clouds developed under the environment with more CCN could have more cloud droplets with smaller sizes, leading to a narrower drop size distribution (DSD). Small cloud droplet sizes and a narrow DSD could result in less collision–coalescence efficiency and then suppress the warm rain processes, known as Albrecht Effect or the second aerosol indirect effect (Albrecht, 1989). However, Albrecht Effect is more relevant to describe the responses of stratocumulus to aerosols, and the ACPI can be cloud-regime dependent (Quaas et al., 2020). For deep convective clouds,



mixed-phase microphysics processes come into play, thus involving more complicated mechanisms to affect precipitation (Tao et al., 2012) How aerosols influence deep convection, which has a higher ability to produce heavy precipitation and a greater probability of causing hazards, is notably the main research target in recent years.

Numerous studies were conducted to explore the impacts of increasing aerosols on convective precipitation. Various types of deep convection were investigated, including squall lines (e.g. Khain et al., 2005; Wang, 2005; Tao et al., 2007;
Lynn et al., 2005; Su et al., 2020; Lebo, 2014; Li et al., 2009; Khain et al., 2004; Lebo and Morrison, 2014), mesoscale convective systems (e.g. Kawecki et al., 2016; Clavner et al., 2018; Zhang et al., 2020), fronts (e.g. Iguchi et al., 2008; Cheng et al., 2010; Liu et al., 2020), and winter cyclones (e.g. Thompson and Eidhammer, 2014; McCoy et al., 2018). The different types of convective systems exhibit inconsistent responses to increasing aerosols/CCN, mainly owing to the different convective structures and organization mechanisms that can significantly feedback to the initial microphysical
perturbations (Khain, 2009; Fan et al., 2016), while the synoptic-scale meteorological conditions modulate which types of convective systems can occur. Since both meteorology and aerosols could influence the development of clouds and precipitation, Stevens and Feingold (2009) stated that the aerosol effects on clouds and precipitation are almost certainly dependent on the weather regimes. Here we generalize the regimes concerning the factors controlling the convective structure to include not only meteorological factors but also topography, land use types, and the other aspects of the
environment.

The deep convective clouds or systems mentioned above are mostly enormous in size with longevity. However, locally and diurnally developed deep convection, that is, afternoon thunderstorms, can still produce extreme precipitation and cause costly hazards. Even if significant synoptic-scale weather forcing is absent, the development of afternoon thunderstorms can still be fueled by the surface heat flux and be affected by the local topography. Since solar heating and corresponding surface
heat flux are directly imposed on the mountain ridges, topography could influence the development of the afternoon thunderstorms. Clear examples can be found in the afternoon thunderstorms and their accompanied diurnal precipitation in Taiwan. Chen et al. (2010) discovered through a case study that the formation and maintenance mechanism of an afternoon thunderstorm system over Snow Mountain Range was related to the lifting of high equivalent potential temperature airflow over the south-western slope. Kuo and Wu (2019) used idealized cloud-resolving model simulations to show that the
confluent flow of sea breezes from two river valleys could determine the location of initiation and the development of afternoon thunderstorms inside Taipei Basin, while the case simulation by Miao and Yang (2020) revealed that the intensified sea breeze and increased moisture transport by the channel effect of the river valley provide favorable dynamic and thermodynamic conditions for more intense convection to develop inside Taipei Basin. Thus, with the tight relationship between afternoon thunderstorms and the local environment, especially topography, we postulate that the influence of
microphysical perturbation on diurnal precipitation through increasing aerosols can be highlighted more evidently in these "orographic-locking" afternoon thunderstorms given similar large-scale weather conditions.

Several studies have introduced the aerosol effects on convective precipitation under different orographic regimes. Seo et al. (2020) showed that the upslope geometry could control the precipitation of shallow convective clouds over a bell-





shaped mountain by conducting two-dimensional idealized simulations. Several simulations concluded that the aerosol
effects suppressed the precipitation of shallow convective clouds in the mountain ranges of the North American Cordillera
(Lynn et al., 2007; Jirak and Cotton, 2006; Givati and Rosenfeld, 2004). Observations from Dominica Experiment field
campaign also revealed that aerosols could have impacts on thermally driven orographic clouds and precipitation (Nugent et
al., 2016). In the studies mentioned above, shallow convection and its resulting precipitation over the topography is the main
focus. However, the aerosol effects on diurnal precipitation induced by deep convection over complex topography remain
insufficiently discerned.

Grabowski and Morrison (2016) showed that the precipitation is strengthened with high CCN concentration based on
the simulation of a diurnal precipitation case during the Large-Scale Biosphere-Atmosphere field campaign over the great
plain of Amazon. However, Grabowski (2018) suggested that the impact of atmospheric environmental perturbations is
comparable to the aerosol effects shown in Grabowski and Morrison (2016). Thus, it is often ambiguous to attribute the
response of deep convection and the resulting precipitation to the aerosol effects. As mentioned previously, the development
of diurnal precipitation in Taiwan is profoundly affected by its complex topography. In this study, we apply the object-based
tracking analyses, which diagnose both the spatial and temporal connectivity of convective systems, to highlight the
convective clouds that are locked by topography and reduce the stochastic features of convection.

Rosenfeld et al. (2008) proposed that deep convection can be invigorated under the environment with more aerosols,
namely the aerosol invigoration effect. Since the warm rain processes are suppressed, more cloud droplets are frozen, and
more latent heat is released above the freezing level. Thus, deep convection would become more intensive and cause more
rainfall under a more polluted environment (Altaratz et al., 2014). The aerosol invigoration effect shows that increasing
aerosols would have a specific influence during various stages of the life cycle of the convective clouds. Therefore, it is
necessary to record the evolution of convection. The adjustment in convective structure and organization is also a crucial
issue of the aerosol effects on deep convection from the dynamical perspective (Su et al., 2020; Lebo and Morrison, 2014;
Fan et al., 2013). The probability distribution of convective features can be altered due to the modulation of the convective
structure by increasing aerosol loading (Su et al., 2020). Therefore, in this study, we specifically focus on the extreme
precipitation and cloud properties of the convective life cycle. Instead of including convection of all stages as an average, the
statistical analyses on extreme convection with the object-based consideration highlight the structural characteristics of
convection modified by increasing aerosols.

The objective of the present study is to investigate how increasing CCN affects the properties of the diurnal
precipitation induced by deep convection under the weak synoptic weather regime over complex topography. We
specifically focus on the precipitating systems produced by orographic-locking processes. Due to the complicated
interactions between convective clouds and their environment, it is challenging to separate the impacts of CCN from the
influence of meteorology on convection merely using observational data (Grabowski, 2018). Thus, we conducted semi-
realistic large-eddy simulations (LESs) with fine temporal and spatial resolutions, highlighting the role of topography on the
evolution of diurnal precipitation. The object-based tracking analyses provide novel and useful insights to the understanding



of the responses of convective systems resulting from increasing CCN. Section 2 presents the model description and the experiment setup. The properties of the diurnal precipitation in Taiwan and the influence of CCN on them over complex

topography are analyzed in Sect. 3, mainly based on the perspective of the precipitating systems. The discussion of the results and the possible extensions that can be accomplished under the semi-realistic LESs framework are presented in Sect. 4, with the summary and conclusion in Sect. 5.

## 2 Model description and semi-realistic experiment setup

### 2.1 Model description

In this study, we use the vector vorticity equation cloud-resolving model (VVM) to simulate the development of the diurnal precipitation over complex topography. VVM was initially developed by Jung and Arakawa (2008), based on the three-dimensional anelastic vorticity equations. In VVM, the horizontal vorticity is predicted, and the vertical velocity is diagnosed using a three-dimensional elliptic equation. The pressure gradient force is eliminated in the equations, and the horizontal buoyancy gradient that drives the vorticity field responds to the surface fluxes directly. Thus, comparing with the

other models using the traditional terrain-following coordinate approach, VVM can better represent local circulation induced by heating differences. The steeper the topography is, the more significant this advantage becomes (Wu et al., 2019). The immersed boundary method is implemented (Chien and Wu, 2016; Wu and Arakawa, 2011) to represent the steep topography in Taiwan. With this representation, mountain waves, orographic precipitation, upslope wind, and other atmospheric phenomenon related to topography can be reasonably simulated without having computational problems. Noah

land surface model (Noah LSM; Chen and Dudhia, 2001; Chen et al., 1996) version 3.4.1 is also coupled to VVM (Wu et al., 2019) and has been applied to evaluate the influence of land–atmosphere interactions on the afternoon thunderstorms on idealized tropical islands (Wu and Chen, 2021).

To investigate the atmospheric processes specifically over Taiwan Island, Wu et al. (2019) developed a framework of VVM with high-resolution Taiwan topography and land use types, named TaiwanVVM. They carried out idealized

simulations of summertime afternoon thunderstorms with realistic Taiwan topography. Hsieh (2019) utilized TaiwanVVM to discuss the effect of local circulation associated with fog formation at Xitou, Nantou County, Taiwan. In contrast to previous studies using TaiwanVVM, this study uses the Predicted Particle Properties (P3; Morrison and Milbrandt, 2015) microphysics scheme implemented by Huang and Wu (2020) to VVM to enable the influences of aerosols to cloud microphysics. The other physical parameterizations used in TaiwanVVM are the Rapid Radiative Transfer Model for GCMs

(RRTMG; Iacono et al., 2008), the flux–profile relationship of Deardorff (1972) to estimate the surface fluxes, and the eddy viscosity and diffusivity coefficients depending on deformation and stability (Shutts and Gray, 1994) as the first-order turbulence closure.

For TaiwanVVM, the horizontal resolution is 500 m. The total vertical layers are 70, and the vertical resolution being 100 m from the sea level up to 3900 m, and a stretched grid above 3900 m up to about 19260 m (Krueger, 1988). The





domain is 512 km × 512 km in size (Fig. 1). To avoid the domain boundary being cut at the edge of complex topography on
Taiwan Island, which might potentially induce problems from the inflow outside the domain, Taiwan Island is placed in the
center of the domain with sufficient area of surrounding seas. To focus on the phenomenon solely related to Taiwan Island,
the topography of adjacent land around Taiwan Island, including several islands, islets, and a part of south-east China, is not
implemented in the model. Although the lateral boundary of TaiwanVVM is doubly periodic, the diurnal convection stays in

the domain under a weak synoptic environment. Other detailed settings of the TaiwanVVM simulation are provided in Table
1.

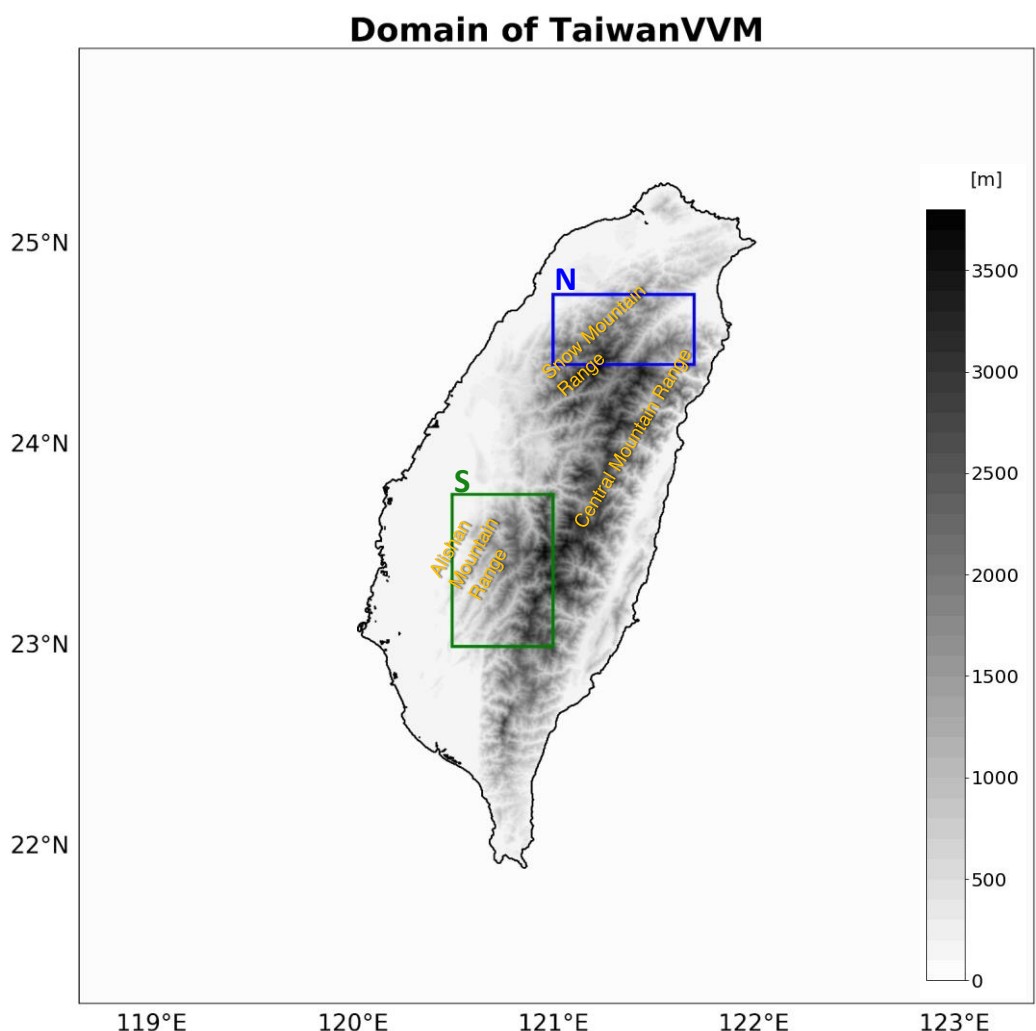

**Figure 1.** The domain of TaiwanVVM with the 500 m resolution topography of Taiwan Island (grey shading). The boxes are
the mountain areas for subsequent statistical analyses: the blue box (N) is the northern mountain area, and the green box (S)
is the southern mountain area.





**Table 1.** The configuration of TaiwanVVM for the semi-realistic simulations.

| Horizontal Resolution | 500 m |
|---|---|
| Vertical Resolution | 100 m under 3900 m<br>Stretch up to 955 m at model top |
| Domain | $1024 \times 1024 \times 70$ grids<br>512 km $\times$ 512 km $\times$ 19260 m |
| Time Step | 10 seconds |
| Simulation Duration | 24 h (00:00–24:00) |
| Lateral Boundary Condition | Double periodic |

## 2.2 Experiment design

A semi-realistic approach is adopted in designing TaiwanVVM simulations. That is, an observed sounding is idealized as the uniform initial condition over the entire domain, similar to Wu et al. (2019). Such an approach is commonly used in LESs (e.g., Grabowski et al., 2006). The direct comparison to the observations of specific cases or events is not the goal of this

study. Instead, the idealization emphasizes the decisive environmental factors that modulate the development of particular convection types. By this semi-realistic approach, interactions among physical processes dominate the evolution of local circulation and convection, which can also interact with the simplified background states in the initial condition. The variability in the background environment is represented by the ensemble approach (mentioned later in Sect. 2.3), and the statistics of the semi-realistic ensemble can be compared with the observed climatological statistics from cases with similar

environments.

To investigate the influence of CCN on the diurnal precipitation over complex topography, we perform experiments with two scenarios of aerosol concentration. In the clean scenario, the aerosol number mixing ratio is fixed at $3 \times 10^8$ kg$^{-1}$ in the entire domain, which is within the range of the clean conditions in the marine environment (Andreae, 2009). Under the normal scenario, on the other hand, the aerosol number mixing ratio increases to $3 \times 10^{10}$ kg$^{-1}$, which lies in the range of the

urban environment of Taipei City, Taiwan (Lin, 2012). In P3, the number of activated CCN ($N_c$) is determined by

$$N_c = \frac{N_a}{2} \left[ 1 - \text{erf}\left( \frac{\ln \frac{s_0}{s}}{\sqrt{2}(1+\beta)\ln \sigma_d} \right), \right. \tag{1}$$

where $s$ is supersaturation, $s_0$ is mean geometric supersaturation, $\beta$ is the soluble fraction of an aerosol particle, and $\sigma_d$ is the dispersion of the dry spectrum. $s_0$ depends on the chemical properties of the soluble part of the dry aerosol, including density, surface tension, van't Hoff factor, osmotic potential, and molecular weight. When $s = s_0$, only half of the total aerosols would be activated as CCN (Khvorostyanov and Curry, 2006; Morrison and Grabowski, 2007; 2008). Thus, the initial

atmospheric conditions are identical, but the aerosol concentration scenarios are different. We can expect that the difference in convection development and convective properties are resulting from the impact of aerosol concentration.


## 2.3 Initial condition

To find appropriate representations for the environment of the diurnal precipitation under weak synoptic-scale weather
forcing in summer, the selecting procedure was carefully designed (Fig. 2). First, by Taiwan Atmospheric Events Database
(TAD; Su et al., 2018), we selected the days with the weak south-westerly flow or weak synoptic weather conditions during
the summers (May to September) between 2005 and 2014. Then, by Central Weather Bureau surface rain gauge observations,
we calculated the average diurnal precipitation cycle of 115 well-functioned weather stations for each day. To find the days
with prominent diurnal precipitation cycle, only the days with precipitation in the afternoon greater than that in the morning,
as well as the diurnal precipitation cycle within two standard deviations were selected. Next, by three-hourly Tropical
Rainfall Measuring Mission Multi-Satellite Precipitation Analysis (3B42) version 7, we chose the days when precipitation
occurred on Taiwan Island, but the coverage of precipitation in the surrounding areas (118.125–123.875º E and 20.625–
26.375º N) was less than 20 %, making sure that the precipitation occurred locally on Taiwan Island. There are 218 days
from the summers between 2005 and 2014 pass the criteria mentioned above. The observed composite precipitation of these
218 days is displayed in Fig. 4a. The precipitation over the mountains is much more intense than that on the plains. The most
significant precipitation hotspot locates around Alishan Mountain Range, which is the green box in Fig. 1 (area S). Another
precipitation hotspot is situated in Snow Mountain Range and the northern tip of Central Mountain Ridge, the blue box in
Fig. 1 (area N), although the observation sites are relatively scarce over there. For these two precipitation hotspots, the
mountain ridges next to the plains have especially more rainfall than the mountain ridges behind them. Finally, we selected
13 days to perform semi-realistic simulations. The selection of these 13 cases largely covers the rainfall variability of the 218
days and serves as the ensemble members representing favorable environments for orographically locked diurnal
precipitation in summer.





**2005-2014 MJJAS**

↓ **1530 days**

**Select Weak Southwesterly (SW) or Weak Synoptic (WS) Weather Types**

**Data**: Taiwan Atmospheric Events Database (TAD)
**Threshold:**
1. weak SW: in the area 110–120º E and 16–22.5º N, (1) 180º < average wind direction < 270º, and (2) average wind speed > 3 m s⁻¹ or > 30% of the area with wind speed > 6 m s⁻¹
2. WS: no synoptic weather type (typhoon, front, southwesterly, or northeasterly) is labelled

↓ **577 days**

**Precipitation Occurs Only in the Afternoon**

**Data**: Central Weather Bureau (CWB) surface rain gauge
**Threshold:**
1. precipitation in the afternoon > precipitation in the morning
2. the diurnal precipitation cycle lies within two standard deviations from the average

↓ **348 days**

**Convection Develops Locally over Taiwan Island**

**Data**: Tropical Rainfall Measuring Mission Multi-Satellite Precipitation Analysis (TRMM/3B42) v.7
**Threshold:**
1. precipitation occurs on Taiwan Island
2. the coverage of precipitation < 20% in the area 118.125–123.875º E and 20.625–26.375º N

↓ **218 days**

**Cases with Favorable Environments for Orographically-Locked Diurnal Precipitation in Summer**

**Figure 2.** The procedure and data of case selection for semi-realistic simulations, aiming to find favorable environments for orographic-locking diurnal precipitation under weak synoptic-scale weather forcing in summer.


The simulations were driven by the simplified Banqiao Station soundings at 08:00 Taiwan Standard Time of these 13 cases. The thermodynamic and dynamic parameters of the initial soundings are shown in Table 2. There is only a small difference among the initial convective available potential energy (CAPE), convective inhibition (CIN), precipitable water (PW), K-index, and mean low-level south-westerly of the 13 simulated cases. High CAPE (at least 900 J kg⁻¹ with the maximum value surpassing 3000 J kg⁻¹), low CIN (mostly less than 50 J kg⁻¹), and high PW (greater than 4.2 cm with the maximum value almost reaching 6.0 cm) indicate that these soundings are conditionally unstable and moist, which are





considered favorable for convection to develop. Low-level south-westerly exists in all 13 soundings, and most of them are south-westerly below 1500 m on average.

**Table 2.** The thermodynamic and dynamic parameters of the 13 initial soundings.

| | Case | Convective Available Potential Energy | Convective Inhibition | Precipitable Water | K-Index | Mean Southwesterly below 1500 m |
|---|---|---|---|---|---|---|
| **SOUTHERN type** | 2007/08/30 | 2092 J kg$^{-1}$ | 11 J kg$^{-1}$ | 4.48 cm | 26 | 2.14 m s$^{-1}$ |
| | 2009/07/07 | 1981 J kg$^{-1}$ | 5 J kg$^{-1}$ | 5.42 cm | 32 | 1.87 m s$^{-1}$ |
| | 2009/08/27 | 2871 J kg$^{-1}$ | 0 J kg$^{-1}$ | 4.65 cm | 24 | 2.88 m s$^{-1}$ |
| | 2010/08/03 | 2306 J kg$^{-1}$ | 64 J kg$^{-1}$ | 5.12 cm | 33 | 3.39 m s$^{-1}$ |
| | 2010/09/12 | 1273 J kg$^{-1}$ | 64 J kg$^{-1}$ | 4.27 cm | 28 | -3.86 m s$^{-1}$ |
| | 2013/08/07 | 3136 J kg$^{-1}$ | 0 J kg$^{-1}$ | 4.76 cm | 26 | 0.83 m s$^{-1}$ |
| **NORTHERN type** | 2006/05/08 | 912 J kg$^{-1}$ | 3 J kg$^{-1}$ | 5.28 cm | 32 | 3.10 m s$^{-1}$ |
| | 2006/07/21 | 2589 J kg$^{-1}$ | 0 J kg$^{-1}$ | 4.36 cm | 20 | 4.68 m s$^{-1}$ |
| | 2010/06/29 | 2237 J kg$^{-1}$ | 32 J kg$^{-1}$ | 5.92 cm | 38 | 1.73 m s$^{-1}$ |
| | 2010/06/30 | 2212 J kg$^{-1}$ | 13 J kg$^{-1}$ | 5.44 cm | 32 | 4.27 m s$^{-1}$ |
| | 2011/08/16 | 1338 J kg$^{-1}$ | 149 J kg$^{-1}$ | 4.51 cm | 30 | 6.51 m s$^{-1}$ |
| | 2012/07/15 | 2824 J kg$^{-1}$ | 0 J kg$^{-1}$ | 4.89 cm | 35 | 4.86 m s$^{-1}$ |
| | 2014/08/25 | 3166 J kg$^{-1}$ | 0 J kg$^{-1}$ | 4.16 cm | 25 | 3.15 m s$^{-1}$ |

Aside from atmospheric conditions, the initial settings of physical parameterizations are listed below. The chemical properties of aerosols are set as ammonium sulfate, and the size distribution of aerosols follows a lognormal size distribution, with a mean size of 0.05 μm (Morrison and Milbrandt, 2015). The initial condition of the ocean and the land is relatively

simple. The surface temperature of the sea and land is prescribed as the temperature of the lowest level of the initial sounding. To drive Noah LSM, land properties are necessary for model inputs. The daily averaged soil moisture over Taiwan Island from the Global Land Data Assimilation System (GLDAS; Rodell et al., 2004) version 2.0 is assigned to the topsoil layers for all land grids in the model. The initial settings of terrain elevation, slope type, land use, green vegetation fraction, and soil texture are the same as in Wu et al. (2019).

**2.4 Object-based tracking algorithm**

Object-based tracking analyses, which combine cloud object connecting and rain cell tracking algorithms, are developed to obtain the statistics related to convective structures and intensity of precipitating systems. Figure 3 is a conceptive example




of the algorithm. The x-z cross-section of two three-dimensional cloud objects is shown at the top of Fig. 3, with their projection to the surface presented beneath. The cloud object connecting is done by the six-connected segmentation method

(Tsai and Wu, 2017). It connects horizontally and vertically adjacent cloudy (cloud liquid water and cloud ice mixing ratio greater than $10^{-4}$) grid boxes as the same cloud object. In this study, only the convective cloud objects, defined by cloud base lower than 0.5 km, cloud depth thicker than 1.0 km, and the center of cloud mass higher than 0.5 km, are analyzed. These criteria are chosen to include the shallow cumulus clouds during the developing stage of convection. For the vertically overlapped cloud objects, the cloud projection on the surface is determined by the lowest cloud object detected bottom-up

from the surface.

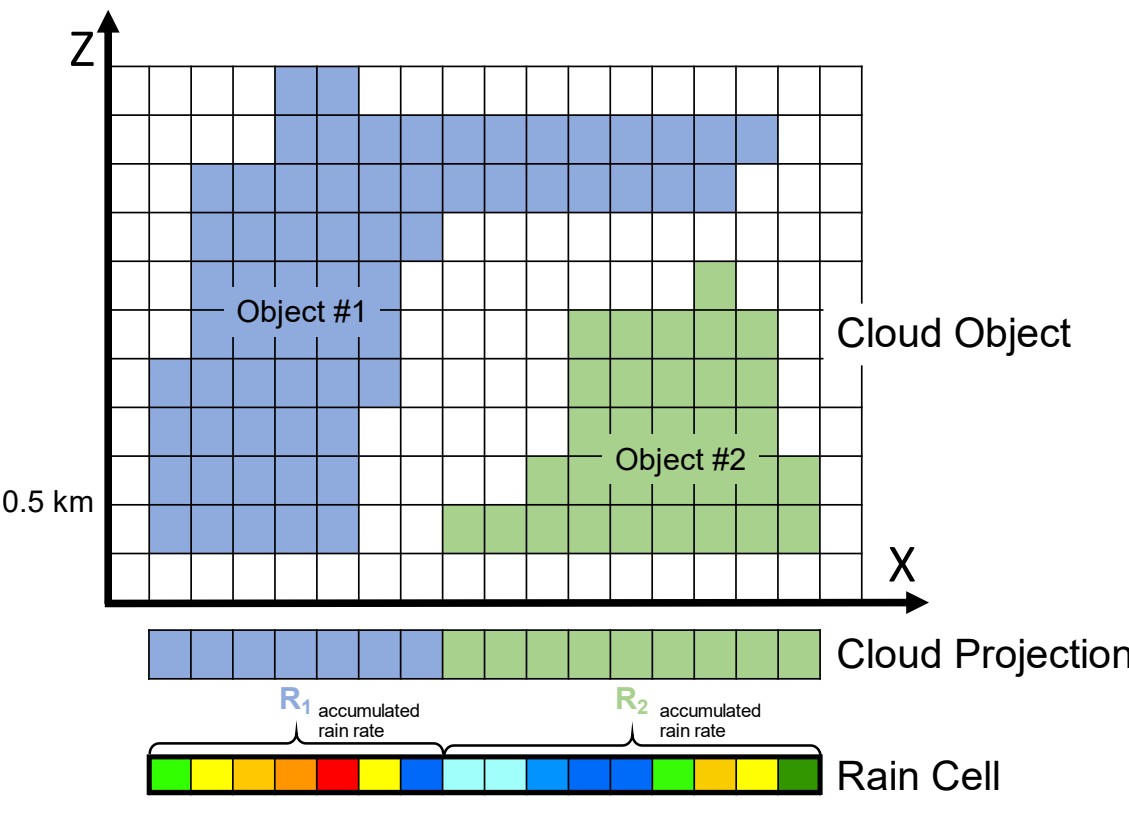

**Figure 3.** The schematic diagram of connecting cloud objects and rain cells, as well as co-locating the cloud objects with the rain cells along the x-z cross-section. Anvil of cloud object #1 (blue cloud) is overlapped with cloud object #2 (green cloud). The cloud projection to the surface is based on the lowest cloud bottom. For a connected rain cell covered by multiple cloud

objects, the co-location is simplified by identifying the cloud object that contributes to highest fractional rainfall. For example, the accumulated rain rate to the rain cell from cloud object #1 is larger than that from cloud object #2, so the rain cell in this diagram would be co-located entirely with cloud object #1.



The bottom of Fig. 3 shows the x-dimension of a two-dimensional rain cell, which is formed by a four-way connection
of rainy grids with a rain rate greater than 5 mm h$^{-1}$. By co-locating the rain cell with the cloud object above, we could
establish the relationship between the precipitation and the convective structure. We simplify the condition of cloud object
overlapping by assuming that the precipitation on the surface is totally contributed by the lowest cloud object. Still, a rain
cell could be covered by multiple cloud objects. For instance, both cloud objects in Fig. 3 partially cover the rain cell. For
this rain cell, the accumulated rain rate covered by cloud object #1 is greater than that of cloud object #2, and we would co-
locate the rain cell fully with cloud object #1. That is, the rain cell would be co-located with the cloud object that contributes
most precipitation to it.

To further evaluate the evolution of precipitating systems, we perform the iterative rain cell tracking (IRT; Moseley et
al., 2013; Moseley et al., 2019). It links the rain cells at each time step and forms the rain tracks, providing a Lagrangian
framework that focuses on the life cycle of the diurnal precipitating systems. By the time connection of the rain cells and the
co-location between rain cells and cloud objects, the life cycle of precipitating systems is established. We can assess the
progression of convective organization and the CCN effect on it.

## 3 Simulation results

In this section, we first present the simulated composite precipitation pattern in Taiwan under the weak synoptic environment.
The composite result of the onset timing of precipitation is also examined, which plays a critical role in the subsequent
convection development, and hence the response of diurnal precipitation to increasing CCN. Lastly, object-based tracking
analyses were carried out to quantify the changes in convective structures of the orographic-locking precipitation systems.

### 3.1 Composite precipitation patterns

Figure 4b demonstrates the composite simulated precipitation in Taiwan of our 13 cases. The simulated precipitation pattern
captures the key features in the observed climatology of the diurnal precipitation under weak synoptic weather in
summertime (Fig. 4a and Lin et al., 2011), particularly the characteristics of more precipitation over the mountains than on
the plains and the location of the two major precipitation hotspots. Two different types of precipitation patterns could be
further distinguished: one with the most significant precipitation hotspot locating in area S (SOUTHERN type hereafter), and
the other one having the highlighted precipitation hotspot in area N (NORTHERN type hereafter). Each type contains 6 and
7 simulated cases, and their composite precipitation patterns are shown in Fig. 4c and d, respectively. Thus, for the
SOUTHERN type, the precipitation in area S occurs under the orographic-locking regime, while the precipitation in area N
occurs under the non-orographic-locking regime. Similarly, for the NORTHERN type, the precipitation in area N occurs
under the orographic-locking regime, while the precipitation in area S occurs under the non-orographic-locking regime. We



expected that a clear ACPI could be discovered under the orographic-locking regime, where the development of the cloud is organized strongly by the topography and is less stochastic and random.


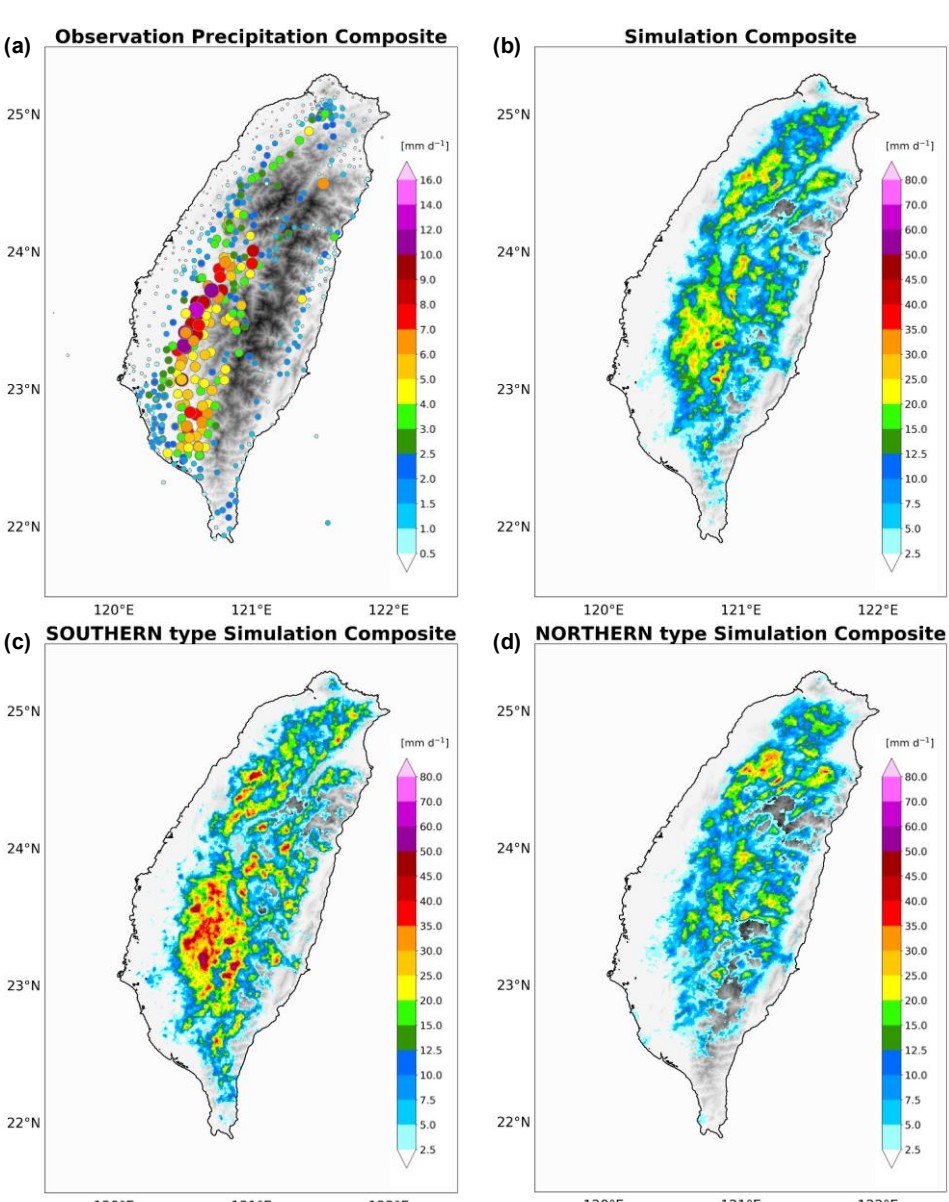

**Figure 4.** The composite daily mean precipitation of **(a)** all 218 weak-synoptic days from Central Weather Bureau rain gauge observations (the sizes of colored dots are scaled with the mean precipitation), **(b)** all 13 semi-realistic simulations, **(c)** six semi-realistic simulations of the SOUTHERN type, and **(d)** seven semi-realistic simulations of the NORTHERN type on Taiwan Island. Grey shadings show the orographic heights (same as Fig. 1).






## 3.2 Initiation time of precipitation

The timing of sufficient solar heating and surface fluxes and the establishment of local circulation determine the initiation time of the diurnal precipitation, which is highly influenced by the topography (Kuo and Wu, 2019). As increasing CCN suppresses the warm rain processes and delays the rain initiation, the changes in the initiation time of precipitation reflect
one of the crucial effects of increasing CCN on diurnal precipitation over complex topography.

To visualize the precipitation timing associated with the topography, a three-dimensional perspective is adopted using VAPOR (Clyne et al., 2007). It is clear to see that, under the clean scenario (Fig. 5a–d), the development of the convective clouds and the initiation time of precipitation is earlier over the mountain ridges and later in the river valleys. The strong buoyancy gradient induced by the heating difference between the mountain ridges and their ambient atmosphere produces
convergent valley breezes that causes the early precipitation over the mountain ridges. As a result, the diurnal precipitation can initiate at noon or even earlier over the mountain ridges. In the river valleys, on the other hand, the diurnal precipitation can be postponed until 16:00 or even later (not shown on the figures), which is possibly caused by the propagation of the precipitating systems.



**Figure 5.** The composite initiation time of precipitation under the clean scenario for the **(a)** SOUTHERN type over S, **(b)** SOUTHERN type over N, **(c)** NORTHERN type over S, and **(d)** NORTHERN type over N. The light blue and dark blue areas represent the initiation time from 10:00 to 12:00 and from 12:00 to 13:00. The area of precipitation initiation delayed for more than 2 hours due to increasing CCN for the **(e)** SOUTHERN type over S, **(f)** SOUTHERN type over N, **(g)** NORTHERN type over S, and **(h)** NORTHERN type over N are also presented. (from ESRI)



Figures 5e–f illustrate the delayed timing of precipitation initiation when CCN concentration increases (i.e., normal
minus clean). For the highlighted areas with significant postponement, the precipitation is usually initiated before 13:00
under the clean scenario. This phenomenon is especially evident for the SOUTHERN type in area S (Fig. 5a and e), where
the western slopes and ridges (pointed out by the yellow arrow) have a precipitation initiation around noon and a significant
rain postponement for about 2 h. We hypothesize that under the orographic-locking regime, increasing CCN could make the
convective clouds have a longer time to develop, leading to stronger intensity and degree of organization. To examine this
hypothesis, we next compare the statistics associated with the convective structures diagnosed by object-based tracking
analyses on diurnal precipitating systems.

### 3.3 Object-based tracking statistics

In this section we apply the object-based tracking analyses, which diagnose both the spatial and temporal connectivity of
convective systems, to highlight the convective clouds that are locked by topography and reduce the stochastic features of
convection. Instead of including convection of all stages as an average, the statistical analyses on the extreme convection
with the object-based consideration feature the structural characteristics of convection modified by increasing aerosols.

Figure 6 presents the counts of occurrence of precipitating systems with the maximum rain rate larger than 100 mm hr$^{-1}$.
The orographic-locking regime (i.e., area S in Fig. 6a and b and area N in Fig. 6c and d) is also the occurrence hotspots of
extreme precipitating systems, and the occurrence is further enhanced with increasing CCN. For the SOUTHERN type in
area S, the counts of the extreme precipitating systems increase by 18, and the hotspot area becomes more extensive in the
normal scenario. For the NORTHERN type in area N, although the occurrence of the extreme precipitating systems remains
similar (about 15), the area of the hotspot also becomes broader when CCN concentration increases. Note that the occurrence
of the extreme precipitating systems over the SOUTHERN type in area S is higher than that over the NORTHERN type in
area N, which implies that orographic-locking is more significant for the SOUTHERN type in area S and that the CCN effect
on extreme precipitating systems is more significant for this regime. For the non-orographic-locking regime (i.e., area N in
Fig. 6a and b and area S in Fig 6c and d), extreme precipitating systems rarely appear, while increasing CCN slightly
elevates their occurrence.





**Figure 6.** The occurrence counts of convective systems with maximum rain rate greater than 100 mm hr$^{-1}$ for **(a)** SOUTHERN/clean, **(b)** SOUTHERN/normal, **(c)** NORTHERN/clean, and **(d)** NORTHERN/normal situations.



In addition to the spatial distribution of the extreme precipitating systems, the effect of increasing CCN can also be identified on the frequency of extreme precipitation, as shown by the probability density function (PDF) of the maximum rain rate of cloud objects (Fig. 7), along with the critical cloud size (CCS). The CCS is defined as the minimum cloud size that can produce the corresponding rain rate. That is, to reach a certain level of rain rate, the clouds have to at least grow into the CCS. For the orographic-locking regime (Fig. 7a and d), the probability of extreme precipitation and CCS is higher than that for the non-orographic-locking regime (Fig. 7b and c). In area S under the clean scenario, the probability of 100 mm h$^{-1}$ rain rate and the corresponding CCS are $4.00\times10^{-6}$ and $1.46\times10^{4}$ km$^3$ for the orographic-locking regime, higher than those for the non-orographic-locking regime ($3.33\times10^{-7}$ and $9.43\times10^{3}$ km$^3$). For the orographic-locking regime, rising CCN leads to a higher probability of extreme precipitation and a larger CCS, particularly for the SOUTHERN type in area S where the probability of 100 mm h$^{-1}$ rain rate and the corresponding CCS are enhanced by $4.13\times10^{-6}$ and $8.48\times10^{3}$ km$^3$, higher than the increase in the NORTHERN type in area N ($8.88\times10^{-7}$ and $8.07\times10^{3}$ km$^3$). For the non-orographic-locking regime, on the other hand, increasing CCN has a negligible effect on either the PDF or the CCS. In area S, when CCN concentration rises, the probability of 100 mm h$^{-1}$ rain rate and the corresponding CCS only increase by $6.10\times10^{-7}$ and $1.71\times10^{3}$ km$^3$ for the NORTHERN type.



**Figure 7.** The probability density functions of the maximum rain rates of the convective cloud objects in **(a)** the SOUTHERN type over S, **(b)** the SOUTHERN type over N, **(c)** the NORTHERN type over S, and **(d)** the NORTHERN type over N. The critical cloud size is defined as the minimum cloud object volume that can produce the corresponding maximum rain rate, presented by the size of the circles. The blue and the red circles represent the results of the clean and the normal scenarios, respectively.

The previous analysis on CCS is the overall statistics on all stages during the lifetime of the diurnal precipitating systems. Next, we further focus on the convective structure and intensity of the mature stage of the diurnal precipitating systems. Figure 8a demonstrates the box-whisker plot of the maximum rain rate during the lifetime of each precipitating system, representing the strength of the precipitation in the mature stage. CCN concentration is more influential on the extreme precipitation of the diurnal precipitating systems for the orographic-locking regime. In area S, the 99th percentile ($P_{99}$) of the maximum rain rate increases by 20.51 mm h$^{-1}$ for the SOUTHERN type when CCN concentration rises, but only



by 8.71 mm h$^{-1}$ for the NORTHERN type. Likewise, in area N, the P$_{99}$ of the maximum rain rate increases by 9.52 mm h$^{-1}$

for the NORTHERN type when CCN concentration rises, but only by 8.60 mm h$^{-1}$ for the SOUTHERN type. The box-whisker plots of the maximum cloud depth and the maximum cloud size during the lifetime of each precipitating system are displayed in Fig. 8b and c, respectively, representing the characteristics of the cloud structures in the mature stage. The P$_{99}$ of the maximum cloud size and the maximum cloud depth for the orographic-locking regime have a more significant response to rising CCN than that for the non-orographic-locking regime. In area S, the P$_{99}$ of the maximum cloud depth increases by

0.30 km for the SOUTHERN type when CCN concentration rises, while it decreases by 0.10 km for the NORTHERN type. The maximum cloud size increases by 8.22×10$^3$ km$^3$ for the SOUTHERN type when CCN concentration rises, but only by 4.03×10$^3$ km$^3$ for the NORTHERN type. In area N, the P$_{99}$ of the maximum cloud depth increases by 0.30 km for both the NORTHERN and the SOUTHERN type when CCN concentration rises. As for the maximum cloud size, the P$_{99}$ of increases by 2.07×10$^4$ km$^3$ for the NORTHERN type when CCN concentration increases, but only by 1.48×10$^4$ km$^3$ for the

SOUTHERN type. Figures 8d and e illustrate the box-whisker plots of the maximum in-cloud vertical velocity and the maximum core ratio during the lifetime of each precipitating system, representing the cloud dynamical features in the mature stage. The core ratio is defined as the fraction of the cloud with the vertical velocity larger than 0.5 m s$^{-1}$, representing the updraft region. Generally, increasing CCN leads to a more intense in-cloud upward motion and a more concentrated core area in the orographic-locking regime. In area S, the mean of the maximum in-cloud vertical velocity increases by 3.49 m s$^{-1}$

for the SOUTHERN type when CCN concentration rises, but only by 1.45 m s$^{-1}$ for the NORTHERN type. The mean of the maximum core ratio decreases by 7.45 % and 2.44 % for the SOUTHERN type and the NORTHERN type, respectively. In area N, the average of the maximum in-cloud vertical velocity only increases by 2.66 m s$^{-1}$ for the NORTHERN type when CCN concentration rises, but by 6.45 m s$^{-1}$ for the SOUTHERN type. The mean of the maximum core ratio also declines by 6.48 % for the NORTHERN type and 3.95 % for the SOUTHERN type.

**Figure 8.** The box-whisker plots of **(a)** the maximum rain rate, **(b)** the maximum cloud depth, **(c)** the maximum cloud size, **(d)** the maximum in-cloud vertical velocity and **(e)** the maximum core ratio during the lifetime of the diurnal precipitating systems. The core ratio is defined as the proportion of the clouds with a vertical velocity greater than 0.5 m·s⁻¹, indicating the ratio of the updraft region. The blue and the red boxes the results of the clean and the normal scenarios, respect. The filled boxes represent the orographic-locking regime, while the hollow boxes represent the non-orographic-locking regime. The dots on the box-whisker plots are the mean values.





In summary, the CCN effect is more significant for the diurnal precipitating systems of the orographic-locking regime. The occurrence of the cloud objects with extreme maximum rain rates doubles. Also, the $P_{99}$ of the maximum rain rate and the maximum cloud size during the lifetime of the diurnal precipitating systems increase by 16.9 % and 6.7 %, respectively.

## 4 Discussion

Stevens and Feingold (2009) pointed out that it is difficult to separate the effect of CCN changes and meteorological perturbations on convective clouds. Also, the variability of convection is so large that it is difficult to make a statistically significant argument of the influence of increasing CCN on them even through numerical modeling experiments (Grabowski, 2018). We show that it is possible to untangle such ambiguity in a more specific condition. This study focuses on the environmental regime of summertime weak synoptic weather over the complex topography of a subtropical island. Under

this environmental regime, the development of convection can be orographically locked so that the effects of increasing CCN can have statistically significant impacts on the convective systems.

The interaction between the convection and the topography-related local circulation is crucial in the mountains. Generally, increasing CCN delays the initiation time of precipitation. Especially for the orographic-locking regime, the significant postponement in the initiation time of precipitation due to increasing CCN prolongs the development of the local

circulation and enables further development of convection. This phenomenon is more evident for the SOUTHERN type in area S, where the development of convection over western slopes and ridges could be linked with local circulation.

Thus, for the orographic-locking regime, the CCN effect on convection becomes significant and manifests on the convective structure and variability, as revealed by the changes in the extreme convective properties. The object-based tracking analyses introduced in this study and the statistics focusing on extreme properties enable us to identify that for the

orographic-locking regime (the top panel of Fig. 9), rising CCN makes the $P_{99}$ of the maximum rain rate, the maximum cloud depth, and the maximum cloud size during the lifetime of the diurnal precipitating systems become much more intense. Meanwhile, the convective clouds of the diurnal precipitating systems generally have a stronger vertical velocity with a more concentrated core area when CCN concentration increases. These results are consistent with the aerosol invigoration effect described by Rosenfeld et al. (2008) during the mature stage of the convective life cycle. For the non-orographic-locking

regime (the bottom panel of Fig. 9), although increasing CCN also leads to a more intense extreme rain rate and convective cloud, the increment of the $P_{99}$ of the maximum rain rate, the maximum cloud depth, and the maximum cloud size during the lifetime of the diurnal precipitating systems is less significant. Fan et al. (2013) reported that aerosols could lead to changes in the macrophysics properties of convection, including cloud top height, cloud depth, and cloud fraction. Our object-based statistics reveal the responses of the detailed morphology and structure of convective systems to aerosols, and the changes in

the probability distribution of the convective properties are evident, showcasing that the object-based tracking analyses of extreme precipitating systems are useful to investigate the responses of orographic-driven diurnal convection to CCN.



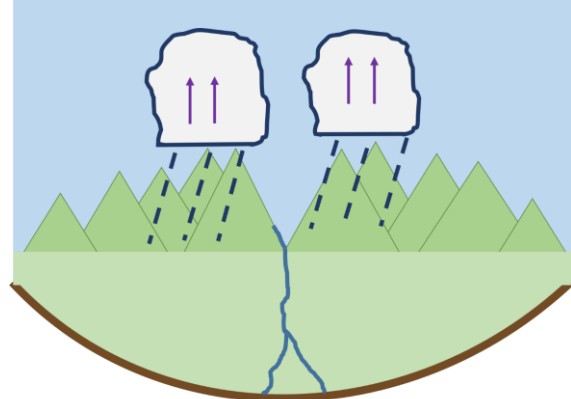

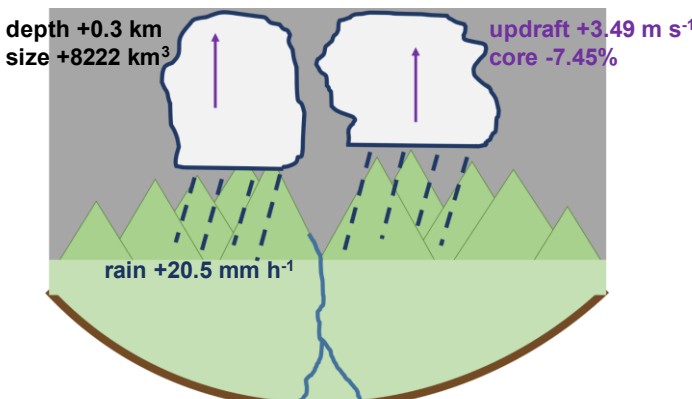

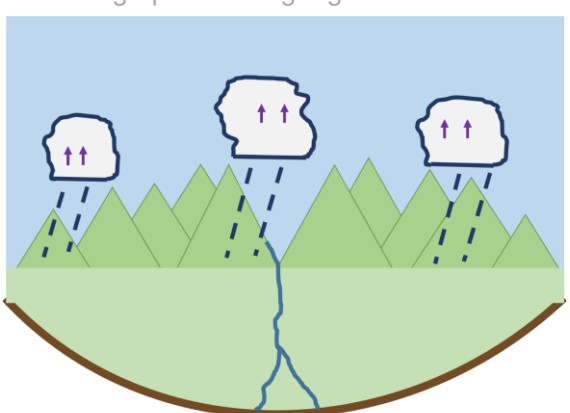

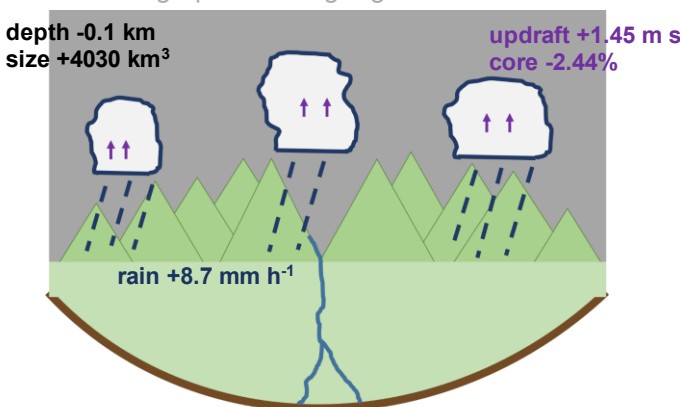

**Figure 9.** The schematic diagram summarizes the influence of CCN on the mature stage of the diurnal precipitating systems over complex topography. The values in the normal scenario are the increment due to increasing CCN in area S for the $P_{99}$ of the maximum rain rate, the $P_{99}$ of the maximum cloud depth, the $P_{99}$ of the maximum cloud size, the mean of the maximum in-cloud vertical velocity, and the mean of the maximum core ratio.

However, why diurnal precipitation in Taiwan shows two distinct patterns is the question that remains to be answered. Since diurnal precipitation is one of the primary water sources for Taiwan in summer, it is critical to understand its relationship with the atmospheric environment and the topography. In addition to the model simulations, high temporal and spatial resolution sounding observation, whose targets are the hotspots of the diurnal precipitation and its upstream surroundings, can produce essential understandings of the fundamentals of diurnal precipitation over complex topography. Through this research, we are confident that TaiwanVVM provides a profound framework to understand the diurnal precipitation over complex topography in Taiwan. Aside from anthropogenic aerosol emissions, global warming and land



use and land cover change are also notable human-induced impacts on the environment. TaiwanVVM can serve as the tool to carry out scenario-based, high-resolution semi-realistic simulations to assess how these factors could alter the characteristics of the diurnal precipitation under summertime weak synoptic weather.

## 5 Summary and conclusion

This study focuses on how CCN concentration affects the properties of the summertime diurnal precipitation under the weak synoptic weather regime over complex topography. Semi-realistic LESs were carried out using TaiwanVVM and driven by idealized observational soundings. Given the same atmospheric environment, the clean and the normal CCN concentration scenarios are simulated. We introduce object-based tracking analyses, aiming to target the aerosol effects on convection of different stages in the life cycle. Two different types of precipitation patterns are identified by their main precipitation

hotspots: the SOUTHERN type and the NORTHERN type. Our results show that for the precipitation types in the corresponding precipitation hotspot (i.e., the orographic-locking regime), the effect of CCN on the diurnal precipitating systems is more significant. For the orographic-locking regime, the precipitation is delayed more significantly due to increasing CCN, which prolongs the development of local circulation and convection. Thus, the convective organization of the diurnal precipitating systems alters. When CCN concentration rises, the diurnal precipitating systems with extreme

maximum rain rates occur more frequently. Also, the maximum precipitation, cloud depth, cloud size during the lifetime of the diurnal precipitating systems become more intense for the normal scenario.

    In conclusion, we argue that CCN could significantly affect the extreme precipitation and cloud features of the diurnal precipitating systems under the summertime weak synoptic weather for the orographic-locking regime. The background weather condition, the topography, and the precipitation type work together to determine the development of the convective

clouds and the effect of CCN on the properties of the convective clouds and the resulting precipitation.





**Data availability**

The observation and data sets were downloaded from the following sources:

a. TRMM 3B42: Tropical Rainfall Measuring Mission (TRMM) (2011), TRMM (TMPA) Rainfall Estimate L3 3 hour
0.25 degree × 0.25 degree V7, Greenbelt, MD, Goddard Earth Sciences Data and Information Services Center (GES DISC), Accessed: [Jan 28, 2021], https://doi.org/10.5067/TRMM/TMPA/3H/7

b. CWB rain gauge and sounding observations: Ministry of Science and Technology & Chinese Culture University, Data Bank for Atmospheric and Hydrologic Research.

c. GLDAS version 2.0 soil moisture: Beaudoing, H. and M. Rodell, NASA/GSFC/HSL (2019), GLDAS Noah Land
Surface Model L4 3 hourly 0.25 × 0.25 degree V2.0, Greenbelt, Maryland, USA, Goddard Earth Sciences Data and Information Services Center (GES DISC), Accessed: [Jan 28, 2021], https://doi.org/10.5067/342OHQM9AK6Q

**Author contribution**

YH Chang and WT Chen designed the experiments and CM Wu performed the simulations. C Moseley developed the tracking algorithm. YH Chang developed the code for analysing observation and model results. CC Wu carried out the 3D
visualization of model outputs. YH Chang and WT Chen prepared the manuscript with contributions from all co-authors.

**Competing interests**

The authors declare that they have no conflict of interest.

**Acknowledgements**

The authors sincerely thank National Center for High-performance Computing (NCHC) for providing the high-performance
computation platform to conduct the simulations. This work is supported by the Ministry of Science and Technology of Taiwan (MOST109-2628-M-002-003-MY3; MOST 107-2111-M-002-010-MY4) and Alexander von Humboldt grant (MOST-AvH 109-2927-I-002-514-).

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
