# Peer review of "Tracking the influence of cloud condensation nuclei on summer diurnal precipitating systems over complex topography in Taiwan"

_Atmospheric Chemistry and Physics, 2021_

## Referee Comment (RC1)

Review Report on "Tracking the influence of cloud condensation nuclei on summer diurnal precipitating systems over complex topography in Taiwan" by Chang et al.

**General comments:**

This manuscript focuses on the effects of increasing aerosols in orographic precipitation under the weak synoptic weather regime. The statistical analysis shows that the impact of CCN on the diurnal precipitating systems are more significant in the orographic-locking regime by postponing the convective initiations, increasing the occurrence of extreme maximum rain rates, and increasing the cloud size. This study successfully untangles the CCN effects and environmental effects in the specific orographic location. I recommend a minor revision.

**Specific comments:**

1. The aerosol number concentration in the polluted scenario is two orders larger than the clean scenario. The polluted scenario comes from Taipei City, and the clean scenario is based on an oceanic situation. I assume these values are applied to the whole domain. Thus, those are the two extreme scenarios. I am curious whether the effects of CCN are also significant if a more realistic aerosol concentration is used. I suggest another simulation with aerosol number concentration as $3 \times 10^9$ kg-1 in the supplementary material.

2. Line 268-270 and other places show the discussions about the circulations. Perhaps Figures showing the circulation can support those explanations better.

3. Figure 5 and the related text discuss the initiation time of precipitation. Based on the text, Figure 5e-h marked the region that CCN has a significant effect in delaying the convection. I think it more informative to use color to represent the delayed time.

4. Line 283-284, "We hypothesis that under the orographic-locking regime, increasing CCN could make the convective clouds have a longer time to develop, leading to stronger intensity and degree of organization." The rest of the manuscript discusses the impacts

of CCN on the precipitation statistics and cloud depth but does not revisit the initiation time and cloud lifetime. Perhaps it is better to reword this sentence.

5. Line 270-300, the effects of CCN are more significant in the orographic-locking regime of the southern type than the northern type. Why?

6. Figure 7, it is better to add grids on the figure to better show orographic-locking regime has a higher maximum rain rate.

---

## Author Comment (AC1)

**Response to Referee Comments on acp-2021-113**

We sincerely thank the referees for their insightful and helpful comments. We have revised the manuscript to address the referee's questions as follows:

1. To increase the statistical sampling, we conducted a larger ensemble to include 30 cases.
2. We modified the analyzing strategy by differentiating the organized regime and the non-organized regime, instead of separating the cases between SOUTHERN and NORTHERN types of geographical areas. **(L274–279)**
3. We have modified the discussion to emphasize that object-based tracking analysis can reduce the variability between different stages of the convective life cycle under the specific regime of summertime weak synoptic weather over complex topography. This enables us to identify the CCN effect on convection in this specific regime is through the mechanism of orographic-induced local circulation. **(L334–342)**

The detailed point-by-point responses to each of the reviewers are enclosed. Both a clean version and a tracked-change version of the revised manuscript are provided. **The line numbers are referred to as the tracked-change version.**

We declare that this manuscript is original, does not infringe upon any other copyright or other proprietary rights of any third party, is not under consideration by another journal, and has not been previously published. The authors confirm that they have reviewed and approved the final version of the manuscript. We declare no conflict of interest, and all the funding sources are explicitly revealed.

**Referee 1**

This manuscript focuses on the effects of increasing aerosols in orographic precipitation under the weak synoptic weather regime. The statistical analysis shows that the impact of CCN on the diurnal precipitating systems are more significant in the orographic-locking regime by postponing the convective initiations, increasing the occurrence of extreme maximum rain rates, and increasing the cloud size. This study successfully untangles the CCN effects and environmental effects in the specific orographic location. I recommend a minor revision.

We sincerely thank the referee for your insightful and helpful comments. We have revised the manuscript to address the referee's questions as follows:
1. To increase the statistical sampling, we conducted a larger ensemble to include 30 cases.
2. We modified the analyzing strategy by differentiating the organized regime and the non-organized regime, instead of separating the cases between SOUTHERN and NORTHERN types of geographical areas. **(L274–279)**
3. We have modified the discussion to emphasize that object-based tracking analysis can reduce the variability between different stages of the convective life cycle under the specific regime of summertime weak synoptic weather over complex topography. This enables us to identify the CCN effect on convection in this specific regime is through the mechanism of orographic-induced local circulation. **(L334–342)**
4. Adding the result of sensitivity tests on aerosol number concentration in Appendix B.
5. Rephrasing the text related to initiation time **(L264–269)**.
6. Revising Fig. 7 by adding grids on the figure.

The detailed point-by-point responses are listed below. **The line numbers are referred to as the tracked-change version.**

1. The aerosol number concentration in the polluted scenario is two orders larger than the clean scenario. The polluted scenario comes from Taipei City, and the clean scenario is based on an oceanic situation. I assume these values are applied to the whole domain. Thus, those are the two extreme scenarios. I am curious whether the effects of CCN are also significant if a more realistic aerosol concentration is used. I suggest another simulation with aerosol number concentration as $3\times10^9$ kg$^{-1}$ in the

supplementary material.

We agree with the reviewer that our experimental design applied strong scenarios in CCN concentrations, and the purpose is to obtain as many statistically significant results as possible. As reviewed in Tao et al. (2012), the effect of increasing aerosols on clouds could approach saturation when the aerosol number concentration is extremely high. Here, we selected four cases (2007/08/30, 2009/08/27, 2010/06/30, and 2012/07/15) to carry out the suggested sensitivity tests in Appendix B, which applied only a 10-times increase in aerosol number concentration ($3\times10^9$ kg$^{-1}$). The counts of occurrence of precipitating systems with the maximum rain rate larger than 100 mm hr-1 in the clean, sensitivity tests (10-fold CCN), normal (100-fold CCN) scenarios are presented in Fig. A. The response in 10-fold CCN experiments is much closer to the 100-fold CCN experiments, indicating that the effects of 100-fold CCN are nearly saturated. The results of clean versus normal scenarios in these four cases are consistent with the analysis of the 30 cases in the main text. To emphasize the overall signal, in the main text we remain presenting the analysis comparing CCN concentration of $3\times10^8$ kg$^{-1}$ (clean scenario) and $3\times10^{10}$ kg$^{-1}$ (normal scenario).

- Tao, W.-K., Chen, J.-P., Li, Z., Wang, C., and Zhang, C.: Impact of aerosols on convective clouds and precipitation, Reviews of Geophysics, 50, https://doi.org/10.1029/2011RG000369, 2012.

[Figure]

**Figure A.** The occurrence counts of convective systems with maximum rain rate greater than 100 mm hr$^{-1}$ when CCN concentration is **(a)** $3\times10^8$ kg$^{-1}$, **(b)** $3\times10^9$ kg$^{-1}$, and **(c)** $3\times10^{10}$ kg$^{-1}$.

2. Line 268-270 and other places show the discussions about the circulations. Perhaps figures showing the circulation can support those explanations better.

The text describes the typical processes during the development of diurnal circulation

over mountainous areas. For clarification, we have included the citation of Houze (2012), which illustrated that solar heating difference between the terrain and their ambient atmosphere produces valley breezes and converges over the mountain ridges. As the air parcels rise above the level of free convection, convective precipitation can be triggered. Without strong synoptic-scale forcing, the diurnal local circulation in Taiwan is driven by the physical processes mentioned above, but the flow over the complicated topography can be highly turbulent and challenging to visualize. Fig B. is an example of the local circulation before the initiation of diurnal precipitation. The wind pattern could be case-dependent, and thus we prefer not to show them in the main text.

- Houze, R. A.: Orographic effects on precipitating clouds, Reviews of Geophysics, 50, https://doi.org/10.1029/2011RG000365, 2012.

[Figure]

**Figure B.** The local circulation at 12:30 in the clean scenario simulation of 2009/07/07.

3. Figure 5 and the related text discuss the initiation time of precipitation. Based on the text, Figure 5e-h marked the region that CCN has a significant effect in delaying the convection. I think it more informative to use color to represent the delayed time.

The 2-dimensional plane view of the delayed timing of precipitation initiation is presented in Fig. C, with all the values shown in colored contours overlayed on the shadings of terrain height. The spatial distribution tightly follows the topography and thus is highly complicated. Figure 5b in the new manuscript plots the identical data, except that only the delayed time over 1.5 hours were shown on the 3-dimensional aerial view. We believe this provides a better visualization to highlight the results that the

delay of the precipitation initiation is especially noticeable over specific mountain ridges where precipitation initiation is the earliest under the clean scenario.

[Figure]

**Figure C.** The delayed timing of precipitation when CCN concentration increases in area S.

4. Line 283-284, "We hypothesis that under the orographic-locking regime, increasing CCN could make the convective clouds have a longer time to develop, leading to stronger intensity and degree of organization." The rest of the manuscript discusses the impacts of CCN on the precipitation statistics and cloud depth but does not revisit the initiation time and cloud lifetime. Perhaps it is better to reword this sentence.

The text has been revised as follows **(L264–269)**:
Thus, we conclude that increasing CCN delays the initiation time of precipitation. This significant delay in precipitation initiation could prevent local circulation from being disrupted by rainfall, which provides the convective clouds a longer time to develop. If

this hypothesis stands, the convection supported by the persisting local circulation could lead to a stronger intensity and higher degree of organization. Therefore, we next compare the statistics associated with the convective structures diagnosed by object-based tracking analyses on diurnal precipitating systems, to examine the relationship between the delay in precipitation initiation and the convective intensity.

5. Line 270-300, the effects of CCN are more significant in the orographic-locking regime of the southern type than the northern type. Why?

To increase the statistical sampling, we conducted a larger ensemble to include 30 cases. Figure 6 presents the counts of occurrence of precipitating systems with the maximum rain rate larger than 100 mm h$^{-1}$. For area S, the counts of the extreme precipitating systems increase significantly from 32 to 52 when CCN concentration rises. As for area N, the number of the extreme precipitating systems due to rising CCN increases significantly from 36 to 37, which is less noticeable than that in area S. Furthermore, the major hotspot in area S remains about the same location, while the major hotspot shifts toward the ridges in the northeast in area N. **(L286–291)**
The organized regime could be discovered in both area S and area N, but with different CCN responses. We postulated that since area S is the direct windward area of south-westerly, and its terrain height increases gradually toward inland, the number of occurrences of extreme precipitating systems increases significantly. As for area N, on the other hand, it is situated in a rather leeward area with relatively equivalent terrain height, so the major hotspot shifts toward the ridges further downwind in the northeast. Under summertime weak synoptic weather with south-westerly, the location and the terrain geometry of these areas could influence the CCN effect on extreme diurnal precipitating systems. **(L375–379)**

6. Figure 7, it is better to add grids on the figure to better show orographic-locking regime has a higher maximum rain rate.

Figure 7 has been revised by adding grids.

**Referee 2**

I have reviewed "Tracking the influence of cloud condensation nuclei on summer diurnal precipitating systems over complex topography in Taiwan" by Chang et al. The manuscript describes a study using a large-area, high-resolution model to investigate the effect of cloud condensation nuclei on orographic precipitation. The use of an ensemble of cases and of an object-tracking algorithm represent a significant advance over the typical case study setup. I recommend publication after my concerns below have been addressed.

We sincerely thank the reviewer for your insightful and helpful comments. We have revised the manuscript to address the reviewer's questions as follows:

1. To increase the statistical sampling, we conducted a larger ensemble to include 30 cases.
2. We modified the analyzing strategy by differentiating the organized regime and the non-organized regime, instead of separating the cases between SOUTHERN and NORTHERN types of geographical areas. **(L274–279)**
3. We have modified the discussion to emphasize that object-based tracking analysis can reduce the variability between different stages of the convective life cycle under the specific regime of summertime weak synoptic weather over complex topography. This enables us to identify the CCN effect on convection in this specific regime is through the mechanism of orographic-induced local circulation. **(L334–342)**
4. Complete the descriptions on several citations.

The detailed point-by-point responses are listed below. **The line numbers are referred to as the tracked-change version.**

1. By and large, I think the focus on a fairly tightly constrained weather system ("orographically locked" precipitation with weak synoptic forcing) and the use of a reasonable-size ensemble of cases are good first steps in the direction of robust conclusions. I would like to see more discussion in the manuscript of remaining uncertainties, however. For example, microphysics uncertainties can still have a very large impact even when the dynamics is fairly well constrained (e.g., White et al., https://doi.org/10.5194/acp-17-12145-2017).

We agree with the referee's comment that microphysics uncertainties can still have a very large impact even when the dynamical environment is fairly well constrained (White et al., 2017). An earlier study using VVM also identify that different

microphysics schemes can lead to a difference in convective structures mainly related to the melting processes at the freezing level (Huang and Wu, 2020). As the focus of this study is to emphasize the importance of local circulation in the CCN influence on convection over complex topography, we will explore the uncertainty related to the microphysics in future studies by conducting mechanism denial experiments to separate the roles of warm rain and cold rain processes in the aerosol invigoration hypothesis (Rosenfeld et al., 2008). **(L368-374)**

- Huang, J.-D., and Wu, C.-M.: Effects of Microphysical Processes on the Precipitation Spectrum in a Strongly Forced Environment, Earth and Space Science, 7, e2020EA001190, https://doi.org/10.1029/2020ea001190, 2020.
- Rosenfeld, D., Lohmann, U., Raga, G. B., Dowd, C. D., Kulmala, M., Fuzzi, S., Reissell, A., and Andreae, M. O.: Flood or Drought: How Do Aerosols Affect Precipitation?, Science, 321, 1309, https://doi.org/10.1126/science.1160606, 2008.
- White, B., Gryspeerdt, E., Stier, P., Morrison, H., Thompson, G., and Kipling, Z.: Uncertainty from the choice of microphysics scheme in convection-permitting models significantly exceeds aerosol effects, Atmos. Chem. Phys., 17, 12145-12175, https://doi.org/10.5194/acp-17-12145-2017, 2017.

2. The authors describe their results as "consistent" with the Rosenfeld et al. (2008) hypothesis and leave it at that. I consider this a missed opportunity to test the Rosenfeld hypothesis -- which is highly controversial -- more deeply. If I am not mistaken, previous work by Grabowski with the same microphysics scheme concluded that the latent heat of freezing is not sufficient to counter the potential energy expenditure of lofting the liquid water mass above the freezing level. Thus, I am a bit surprised by the authors' conclusion, and I think they are doing themselves a disservice by not digging deeper. For example, if they disable freezing processes, do they still obtain the same invigoration signal (i.e., would warm rain invigoration suffice as a mechanism to explain the model behavior)? One of the advantages of modeling studies over observations is that they allow these types of process denial studies.

Grabowski (2018) showed that the variability of convection is so large that it is difficult to make a statistically significant argument of the influence of increasing CCN on them even through numerical modeling experiments. However, we proposed that it is possible to untangle such ambiguity in a more specific condition in terms of weather, topography, and convective life cycle. This study focuses on the environmental regime

of summertime weak synoptic weather over the complex topography of a subtropical island. Under this environmental regime, the development of convection can be orographically locked. Thus, the convection would become more organized with a higher chance of extreme precipitation. Rosenfeld et al. (2008) also showed that variability exists among different stages of the convective life cycle. Thus, we conduct object-based tracking analysis and focus on the statistical features of the mature stage of the diurnal precipitating systems. The result shows that the CCN effect on convection becomes significant and manifests on the convective structure and variability for the organized regime. Although the result of more intense precipitation and convection organization is similar to the aerosol invigoration effect in Rosenfeld et al. (2008), the mechanism proposed in this study is through the sustainment of local circulation, which is induced by the postponement of rain initiation. We have included the above description in the discussion section. **(L334–342)** We agree with the referee's comment that the experiments on the microphysical processes can be informative to further identify the sensitivity of current results to various microphysical mechanisms. However, the focus of this study is to stress a delicately selected regime of summertime weak synoptic weather over the complex topography, and on the mature stage of the convective life cycle. Therefore, we will explore the experiments on microphysics in future studies.

3. This might be more an indication that I am braindead than anything wrong with the manuscript, but I did not follow the northern/southern locked/not locked argument. I can see that there are two regimes, one where the strongest precipitation occurs in the northern region and one where it occurs in the southern region (though I would be really curious how sharp this distinction is in the two ensemble members closest to the regime split). But I don't understand why precipitation in region N during a southern case can't still be orographically locked.

We thank the referee's comment. After re-examining our simulation results, the aggregated convection induced by topography can be discovered in both areas S and N. Therefore, we no longer address the separation of SOUTHERN type and NORTHERN type of geographical area. Instead, we identify the organized regime and the non-organized regime by the size of the convective systems. For a simulated case under the clean scenario in a precipitation hotspots area, once the $75^{th}$ percentile of the maximum cloud size during the lifetime of the diurnal precipitating systems is greater than $3\times10^4$ km$^3$, the area of the case would be considered as the organized regime. The classification of the organized regime in areas S and N among the 30 simulations are

listed in Table A1 in Appendix A. The effect of CCN can be clearly identified in the organized regime. **(L274–279)**

4. I think the authors could make a stronger case for their method by describing the advantage that object tracking conveys. For example, what conclusions does their object tracking analysis permit that they could not have drawn from an old-school, area-mean analysis?

In this study, the application of object-based tracking analyses enables us to identify the organized regime, as well as to focus on the mature stage of the convective life cycle. This reduces the variability of convection and highlights the CCN effect. In a direct areal-mean analysis, the results would be dominated by weak precipitating systems that occur more frequently, and the change in the extreme would not be revealed as clearly. **(L350-354)**

Minor Comments:
1. Perhaps I missed it, but how interactive are the CCN? I.e., are they depleted by wet scavenging?

The text has been revised as follows **(L118–121)**:
In contrast to previous studies using TaiwanVVM, this study uses the Predicted Particle Properties (P3; Morrison and Milbrandt, 2015) microphysics scheme, implemented by Huang and Wu (2020) to VVM, to enable the influences of aerosols to cloud microphysics, while the aerosols are not scavenged by precipitation.

2. L29: Albrecht (1989) actually discusses both stratocu and shallow cu. Perhaps replace "stratocumulus" with "warm clouds"?

The text has been revised as follows **(L26–27)**:
However, Albrecht Effect is more relevant to describe the responses of warm clouds to aerosols, and the ACPI can be cloud-regime dependent (Mulmenstadt and Feingold, 2018).

3. L29: The Quaas et al. (2020) review is mostly concerned with the Twomey effect,

not precipitation. Other reviews, perhaps Mulmenstadt and Feingold (https://doi.org/10.1007/s40641-018-0089-y), provide a more general discussion of regime dependence in ACI.

The text has been revised as follows **(L26–27)**:
However, Albrecht Effect is more relevant to describe the responses of warm clouds to aerosols, and the ACPI can be cloud-regime dependent (Mulmenstadt and Feingold, 2018).